# ENABLING MODEL PARALLELISM FOR NEURAL NETWORKS BASED ON DECOUPLED SUPERVISED CONTRASTIVE LEARNING

## ABSTRACT

End-to-end backpropagation (BP) is the current standard for training deep neural networks. However, as networks become deeper, BP becomes inefficient for various reasons. This paper introduces a new methodology that decouples BP, transforming a long gradient flow into multiple short ones. This design enables the *simultaneous* computation of parameter gradients in different layers so as to realize better model parallelism. Thorough experiments are presented to demonstrate the efficiency and effectiveness of our model compared to BP, Early Exit, GPipe, and associated learning (AL), a state-of-the-art methodology for backpropagation decoupling. The experimental code is released for reproducibility at `https://anonymous.4open.science/r/SCPL-802C/`

## 1 INTRODUCTION

Large neural networks have become prominent. However, training large neural networks poses several challenges, with one of the difficulties being the issue of *backward locking*: when dividing a large neural network into segments processed across multiple GPUs, the chain rule restricts each GPU's layers to wait for gradient information from layers in other GPUs closer to the target before proceeding with their gradient computations. Backward locking severely limits the potential speedup that parallel processing could offer, as the sequential nature of gradient calculations becomes a bottleneck of the training process.

This paper proposes a simple yet innovative methodology, Supervised Contrastive Parallel Learning (SCPL), to address the issue of backward locking. As a result, SCPL realizes *model parallelism*, which not only partitions a large neural network into segments processed across multiple computing units (e.g., GPUs) but also enables parallel processing on these devices. This is similar but different from most of today's model parallelism tools or packages (e.g., the definitions used by Amazon SageMaker[1]), which allocates different model components to different GPUs but not necessarily running these GPUs simultaneously.

SCPL decouples the long gradient flow of a deep neural network by leveraging supervised contrastive learning (SCL). The forward path transforms an input $x$ into the corresponding prediction $\hat{y}$ as a usual neural network in this design. However, the gradient flow on the backward path is blocked between different components. Instead of using a global objective, SCPL assigns a local objective to each component and forces each gradient flow to remain within one component. When allocating these local objectives to different GPUs, SCPL can compute the local gradients without waiting for the gradients in the neighboring layers.

We conduct experiments on multiple open datasets, including computer vision and natural language processing tasks, using famous network structures such as the vanilla convolutional neural network, VGG, ResNet, LSTM, and Transformer. Our results show that SCPL increases training throughput while maintaining comparable test accuracy compared to models trained via backpropagation (BP), Early Exit, and Associated Learning (AL) (Wu et al., 2022; Kao & Chen, 2021), a state-of-the-art

---

[1]`https://docs.aws.amazon.com/sagemaker/latest/dg/model-parallel-intro.html`

Table 1: A comparison of the properties of the representative related models ($H$: number of layers/-components)

| Method | Supervised? | Parallel model training? | Length of gradient flows |
|---|---|---|---|
| BP (Rumelhart et al., 1986) | Y | N | $O(H)$ |
| AL (Wu et al., 2022) | Y | Y (but not implemented) | $O(1)$ |
| GPipe (Huang et al., 2019) | Y | Y | $O(H)$ |
| LoCo (Xiong et al., 2020) | N | Y (but not implemented) | $O(1)$ |
| GIM (Löwe et al., 2019) | N | Y (but not implemented) | $O(1)$ |
| SCPL (ours) | Y | Y | $O(1)$ |

methodology for decoupling BP. We released the code and Docker images with a step-by-step guide for reproducibility.

The rest of the paper is organized as follows. In Section 2, we review previous works on model parallelism. In Section 3, we introduce SCPL and its properties. Section 4 compares SCPL with BP, Early Exit, and AL with respect to their training time and test accuracies. We conclude our contribution in Section 5.

## 2 RELATED WORK

### 2.1 DATA PARALLELISM VS. MODEL PARALLELISM

Data parallelism (DP) and model parallelism (MP) are two strategies employed in distributed deep learning to mitigate the considerable training time required for neural network models. DP involves distributing batches of training data across multiple devices or processors; each device computes the gradients independently. Devices usually synchronize gradients at the end of every iteration or multiple iterations (Shallue et al., 2018; Li et al., 2020). DP is well-suited for scenarios where the model can fit within the memory of each device.

On the other hand, MP tackles the issue of training models that are too large to fit into the memory of a single device. MP divides the model into segments that are processed on different devices. However, the interdependence of gradient calculations across layers complicates the parallelization process. Each device needs the gradients from other devices to proceed with its gradient calculations, creating a synchronization bottleneck that limits the potential parallelism. As a result, naïve model parallelism (which will be called "NMP" below) results in poor training efficiency.

### 2.2 MODEL PARALLELISM STRATEGIES

A core challenge in model parallelism, backward locking, stems from the inherent characteristics of backpropagation (BP). Therefore, the various methods that attempt to find alternative BP methods could potentially achieve model parallelism. Studies in this line include target propagation (TP) (Lee et al., 2015; Meulemans et al., 2020; Manchev & Spratling, 2020; Bengio, 2014), gradient prediction (Jaderberg et al., 2017), and local objective assignments (Wu et al., 2022; Kao & Chen, 2021). Although many of these methods do not require gradient computation in training, most still need to update the parameters layer-by-layer, so it is challenging to realize model parallelism. Associated learning (AL) is one of the very few methods capable of simultaneously updating parameters in different layers. However, the authors of BP only released a sequential implementation, making it hard to experiment and verify AL's parallelization capacity.

NMP has a low training efficiency, but MP's training efficiency can be improved by pipelining, which overlaps GPU computations and minimizes idle time between different stages of computation. The most representative model in this line is Google's GPipe (Huang et al., 2019), which separates mini-batches into micro-batches to pipeline the forward and backward pass. However, GPipe still relies on the chain rule for gradient computation, so bubbles are inevitable, and the length of a gradient flow is $O(H)$, identical to BP. Some following works, e.g., PipeDream Narayanan et al. (2019), further overlap forward and backward passes, thereby reducing synchronization overhead.

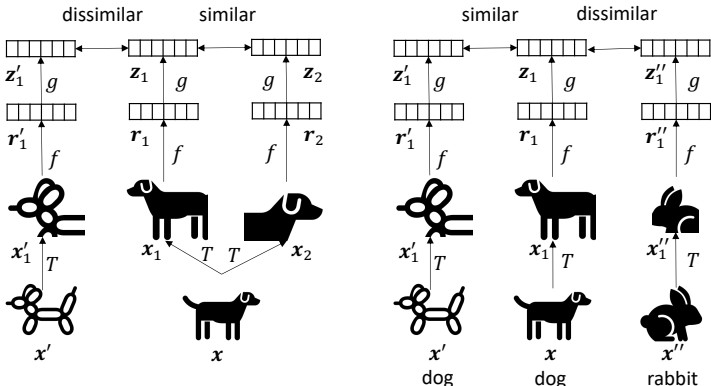

Figure 1: An illustration of contrastive learning (CL) and supervised contrastive learning (SCL). CL regards an anchor image's augmented images as positive pairs (e.g., $x_1$ and $x_2$ above) and regards the anchor image's augmented image to all non-augmented images as negative pairs (e.g., $x_1$ and $x_1'$ on the left). SCL regards augmented images as positive pairs if they have the same label (e.g., $x_1$ and $x_1'$ on the right); a pair of augmented images is a negative pair if their labels are different (e.g., $x_1$ and $x_1''$).

However, the lack of global gradient synchronization leads to challenges in maintaining accurate gradient updates, and thus, the final model tends to be underfitting.

Some studies used self-supervised learning to decouple BP, e.g., Greedy InfoMax (GIM) (Löwe et al., 2019) and LoCo (Xiong et al., 2020). While these studies localize the losses, they mainly focus on improving model accuracy and modularity but pay little attention to model parallelism. Additionally, these works are primarily unsupervised (self-supervised), so their results are not directly comparable to those of our work.

Our proposed SCPL is motivated by studies in pipeline model parallelism (e.g., AL and GPipe) and BP decoupling using self-supervised learning (e.g., GIM and LoCo). However, they are different in several ways. Referring to Table 1, GIM and LoCo are unsupervised, and their released implementation does not support model parallelism. Therefore, although these methods decouple gradient flows and the length of each gradient flow is $O(1)$ (i.e., its length is irrelevant to the number of layers), they cannot be directly compared to SCPL. BP, AL, and GPipe are supervised. However, BP does not support model parallelism due to backward locking, and the length of BP's gradient flow is linearly proportional to the number of layers, $O(H)$. Although AL's design addresses backward locking, thus enabling model parallelism, the released implementation is sequential. Converting AL's sequential implementation to support model parallelism still requires significant engineering effort. GPipe remains suffering from backward locking, so the bubbles are unavoidable. Integration of GPipe and SCPL is possible (see Appendix A.5 for details). However, we leave this part for future work.

## 3 METHODOLOGY

### 3.1 PRELIMINARIES: CONTRASTIVE LEARNING AND SUPERVISED CONTRASTIVE LEARNING

Contrastive learning (CL) is a self-supervised technique for learning the representations of objects. Referring to the left of Figure 1, given an image $x$, CL involves generating different views (i.e., $x_1$ and $x_2$) through the same family of data augmentations $T$. The generated views ($x_1$ and $x_2$) are further transformed using an encoder $f$ and a projection head $g$ to minimize the contrastive loss between the output vectors (i.e., $z_1$ and $z_2$). After training, the projection head $g$ is disregarded, and only the encoder $f$ is used to generate the representations of the images (Chen et al., 2020). In other words, given an anchor image $x$, CL regards $x$'s augmented images as positive instances and all other images as negative instances, and positive pairs should be similar after encoding and projection.

SCL extends CL from a self-supervised setting to a fully supervised setting. Therefore, the training data for SCL consist of not only the training features but also the labels. Referring to the right of

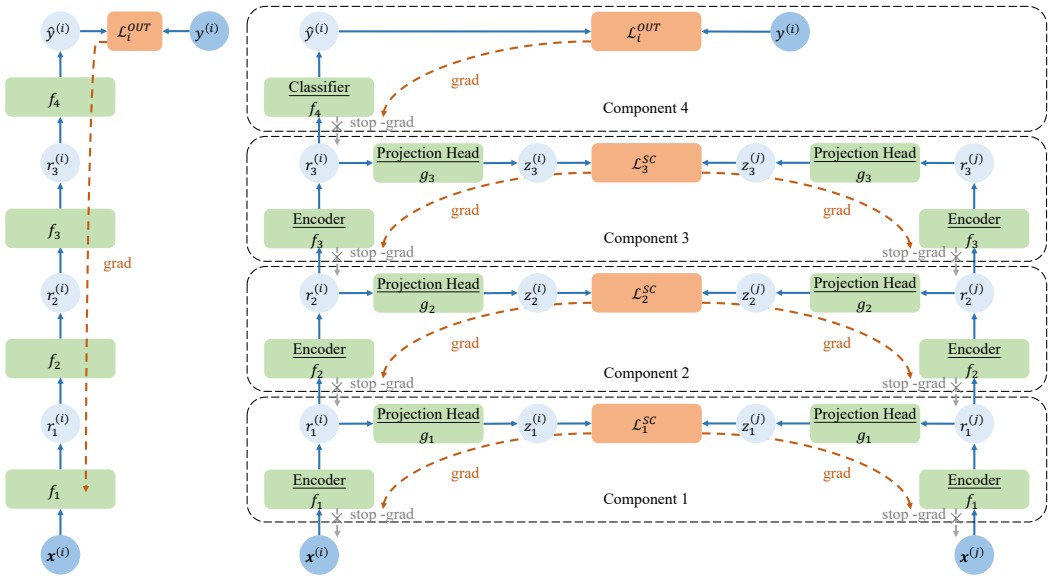

Figure 2: An example neural network with 3 hidden layers and its corresponding SCPL network. Solid blue arrows correspond to forward paths, red dashed arrows correspond to backward paths, green boxes denote the parameters (functions), and orange boxes represent the loss functions. The gradient flows are blocked between neighboring blocks for SCPL.

Figure 1, given an anchor image $\boldsymbol{x}$ with label $c$, the positive instances include the augmented images of $\boldsymbol{x}$ and other images (along with their augmented images) of label $c$ in the same batch, whereas all other images in the same batch are regarded as negative instances (Khosla et al., 2020).

## 3.2 Decoupling end-to-end backpropagation via supervised contrastive learning

This section presents SCPL, which leverages the supervised contrastive loss to split one long gradient flow in a deep neural network into multiple shorter ones.

Let us first consider a standard neural network with 3 hidden layers as an example. As shown in the left of Figure 2, $\boldsymbol{x}^{(i)}$ refers to an input image $i$, and the function $f_\ell$ ($\ell = 1, \ldots, 4$) transforms $\boldsymbol{r}_{\ell-1}^{(i)}$ into $\boldsymbol{r}_\ell^{(i)}$ (under the assumptions that $\boldsymbol{x}^{(i)} = \boldsymbol{r}_0^{(i)}$ and the predicted class $\hat{y}^{(i)} = \boldsymbol{r}_4^{(i)}$). Depending on the network architecture, the functions $f_\ell$ could be various neural network layers, such as fully connected layers, convolutional layers, pooling layers, or residual blocks. The objective $\mathcal{L}^{OUT}$ is determined by the task type. For example, a classification task typically uses the cross-entropy loss between the predicted $\hat{y}^{(i)}$ and the ground-truth class $y^{(i)}$ as $\mathcal{L}^{OUT}$. We use backpropagation to obtain $\partial \mathcal{L}^{OUT} / \partial \theta_{f_\ell}$ for each layer $\ell$, where $\theta_{f_\ell}$ represents the parameters of function $f_\ell$. Once the gradients are obtained, we can use gradient-based optimization strategies, e.g., gradient descent, to update the parameter values. Given a neural network with $H$ hidden layers, the longest gradient flow is constructed as a product of $H + 2$ local gradients. For example, to obtain $\partial \mathcal{L}^{OUT} / \theta_{f_1}$ in a network with 3 hidden layers (as shown in the left of Figure 2), we need the following:

$$\frac{\partial \mathcal{L}^{OUT}}{\partial \theta_{f_1}} = \frac{\partial \mathcal{L}^{OUT}}{\partial \hat{y}^{(i)}} \times \frac{\partial \hat{y}^{(i)}}{\partial r_3^{(i)}} \times \frac{\partial r_3^{(i)}}{\partial r_2^{(i)}} \times \frac{\partial r_2^{(i)}}{\partial r_1^{(i)}} \times \frac{\partial r_1^{(i)}}{\partial \theta_{f_1}}. \tag{1}$$

The number of terms of this product grows linearly with the depth of the network. Therefore, as a network becomes deeper, its long gradient flow may cause optimization and performance issues, as discussed in Section 1.

We use Figure 2 to illustrate our strategy of cutting a long gradient flow into several local gradients for a neural network with 3 hidden layers. Let $r_0^{(i)}$ (i.e., $x^{(i)}$) and $r_0^{(j)}$ (i.e., $x^{(j)}$) be two image views in the same batch ($r_0^{(i)}$ and $r_0^{(j)}$ may or may not be augmented images, i.e., views, from the same image). We use $f_1$ to transform each of them, obtaining $r_1^{(i)}$ and $r_1^{(j)}$, and further use the function $g_1$ to convert them into $z_1^{(i)}$ and $z_1^{(j)}$, respectively. The functions $f_1$ and $g_1$ can be considered as the encoder and the projection head, respectively, in CL (refer to Figure 1). We repeat the same process for each hidden layer $\ell$ to form the corresponding component $\ell$. If $x^{(i)}$ and $x^{(j)}$ are two different views of the same image or if $y^{(i)}$ (the label of $x^{(i)}$) is the same as $y^{(j)}$ (the label of $x^{(j)}$), then we should ensure that $z_\ell^{(i)}$ is close to $z_\ell^{(j)}$ for all $\ell$. Otherwise, we should increase their distance. Eventually, we define the local supervised contrastive loss $L_\ell^{SC}$ for batch $B$ in layer $\ell$ as below.

$$\mathcal{L}_\ell^{SC}(B) = \sum_{\forall i \in B} \frac{-1}{|P(i)|} \sum_{\forall p \in P(i)} \log \frac{\exp\left(z_\ell^{(i)} \cdot z_\ell^{(p)}/\tau\right)}{\sum_{\substack{\forall j \in B \\ j \neq i}} \exp\left(z_\ell^{(i)} \cdot z_\ell^{(j)}/\tau\right)}, \tag{2}$$

where $B = \{1, 2, \ldots, b\}$ represents a batch of multiview images ($b = 2N$ if using data augmentation and $b = N$ otherwise), $P(i)$ is the set of all positive samples for an image $i$, $\tau$ is a hyperparameter, and $I(j \neq i) \in \{0, 1\}$ is an indicator function that returns 1 if $j \neq i$ and 0 otherwise.

Ultimately, the global objective function of a batch $B$ is an accumulation of the local supervised contrastive losses and the losses in the output layer (the distance between $\hat{y}^{(i)}$ and $y^{(i)}$):

$$\mathcal{L}(B) = \sum_{\ell=1}^{H} \mathcal{L}_\ell^{SC}(B) + \sum_{\forall i \in B} \mathcal{L}_i^{OUT}, \tag{3}$$

where $H$ is the number of hidden layers, and $\mathcal{L}_i^{OUT}$ is the $i$th loss in the output layer (refer to right of Figure 2).

Detailed structures and hyperparameters are given in Appendix A.1. The computation of $\mathcal{L}_\ell^{SC}$ and the pseudocode of SCPL for a 3-layer vanilla ConvNet is given in Algorithm 1 and Algorithm 2, respectively, in Appendix A.6.

### 3.3 FORWARD PATH, BACKWARD PATH, AND INFERENCE FUNCTION

For a regular neural network (e.g., left of Figure 2), the forward path and the inference function are identical, and the backward path is simply obtained by inverting the direction of the forward path. However, SCPL is different because we divide the objective into several local ones. Consequently, we have multiple short local forward paths, multiple short local backward paths, and one global inference path; the inference path and the forward paths are no longer identical in SCPL.

During training, each component $\ell$ has its own forward and backward paths. Taking the SCPL network in Figure 2 as an example, the forward path of component $\ell$ transforms each $r_{\ell-1}^{(i)}$ into $r_\ell^{(i)}$ via the local encoder $f_\ell$ and further transforms each $r_\ell^{(i)}$ into $z_\ell^{(i)}$ via the local projection head $g_\ell$. On the backward path, each hidden layer computes $\partial \mathcal{L}_\ell^{SC}/\partial \theta_{g_\ell}$ and $\partial \mathcal{L}_\ell^{SC}/\partial \theta_{f_\ell}$ based on the chain rule and updates the parameters by gradient-based optimization strategies. We block the gradient flow between each component. As a result, each gradient flow remains within one component. Equation 4 and Equation 5 show these local gradient flows.

$$\frac{\partial \mathcal{L}_\ell^{SC}}{\partial \theta_{g_\ell}} = \frac{\partial \mathcal{L}_\ell^{SC}}{\partial z_\ell^{(i)}} \times \frac{\partial z_\ell^{(i)}}{\partial \theta_{g_\ell}}. \tag{4}$$

$$\frac{\partial \mathcal{L}_\ell^{SC}}{\partial \theta_{f_\ell}} = \frac{\partial \mathcal{L}_\ell^{SC}}{\partial z_\ell^{(i)}} \times \frac{\partial z_\ell^{(i)}}{\partial r_\ell^{(i)}} \times \frac{\partial r_\ell^{(i)}}{\partial \theta_{f_\ell}}. \tag{5}$$

**Standard BP**

| Device No. | Stage | | | | | | | | | | | | | | | | |
|---|---|---|---|---|---|---|---|---|---|---|---|---|---|---|---|---|---|
| GPU0 | FW1 | FW2 | FW3 | FW4 | LOSS | BW4 | BW3 | | BW2 | | | | BW1 | | | | UP |
| Time point | $t_1$ | $t_2$ | $t_3$ | $t_4$ | $t_5$ | $t_6$ | $t_7$ | $t_8$ | $t_9$ | $t_{10}$ | $t_{11}$ | $t_{12}$ | $t_{13}$ | $t_{14}$ | $t_{15}$ | $t_{16}$ | $t_{17}$ |

**NMP**

| Device No. | Stage | | | | | | | | | | | | | | | | |
|---|---|---|---|---|---|---|---|---|---|---|---|---|---|---|---|---|---|
| GPU0 | FW1 | | | | | | | | | | | | BW1 | | | | UP |
| GPU1 | | FW2 | | | | | | | BW2 | | | | | | | | UP |
| GPU2 | | | FW3 | | | | BW3 | | | | | | | | | | UP |
| GPU3 | | | | FW4 | LOSS | BW4 | | | | | | | | | | | UP |
| Time point | $t_1$ | $t_2$ | $t_3$ | $t_4$ | $t_5$ | $t_6$ | $t_7$ | $t_8$ | $t_9$ | $t_{10}$ | $t_{11}$ | $t_{12}$ | $t_{13}$ | $t_{14}$ | $t_{15}$ | $t_{16}$ | $t_{17}$ |

**SCPL**

| Device No. | Stage | | | | | | | |
|---|---|---|---|---|---|---|---|---|
| GPU0 | FW1 | LOSS | BW1 | | | | UP | |
| GPU1 | | FW2 | LOSS | BW2 | | | UP | |
| GPU2 | | | FW3 | LOSS | BW3 | | UP | |
| GPU3 | | | | FW4 | LOSS | BW4 | UP | |
| Time point | $t_1$ | $t_2$ | $t_3$ | $t_4$ | $t_5$ | $t_6$ | $t_7$ | $t_8$ |

FW$i$: forward for layer $i$
LOSS: compute loss
BW$i$: backward for layer $i$
UP: update parameter values

Figure 3: An illustrating example to compare the GPU usage of one iteration for standard BP, NMP, and SCPL. The true GPU utilization is shown in Figure 4 in Appendix A.2

Eventually, even if we construct a deep neural network, the cost of computing each $\partial \mathcal{L}^{SC}/\partial \theta_{f_\ell}$ and each $\partial \mathcal{L}^{SC}/\partial \theta_{g_\ell}$ remains constant (i.e., $O(1)$). Additionally, the gradient flow in the output layer is also short: we simply compute $\partial \mathcal{L}_k^{out}/\partial \theta_{f_{H+1}}$ (where $H$ is the number of hidden layers).

In the inference (prediction) phase, we need only the encoders $f_\ell$ but not the projection heads $g_\ell$, as shown by Equation 6:

$$\hat{y}^{(i)} = f_{H+1} \circ f_H \circ \ldots \circ f_2 \circ f_1(\boldsymbol{x}^{(i)}), \tag{6}$$

where $\circ$ is the function composition operator ($H = 3$ for the example illustrated in Figure 2).

Although our proposed method (e.g., right of Figure 2) involves more parameters than a standard neural network structure (e.g., left of Figure 2) during training, they have the same number of parameters during inference because both of them use only the functions $f_\ell$. Therefore, they have the same hypothesis space. The parameters that participate in the inference phase ($\theta_{f_\ell}$-s) are called the *effective parameters*. The parameters used during training but not during inference ($\theta_{g_\ell}$-s) are called the *affiliated parameters*. As a bonus effect, having affiliated (redundant) parameters sometimes helps optimization (Arora et al., 2018; Chen & Chen, 2020).

### 3.4 PARALLELIZATION VIA PIPELINING

Since each component has its local objective, we can parallelize the training procedure via pipelining. We use the network illustrated in Figure 2 as an example. Referring to Figure 3, the top subfigure shows a standard learning iteration using a single GPU. The middle subfigure illustrates NMP by segmenting the model into 4 components and assigning each part to 1 GPU (we ignore the communication cost between GPUs). However, due to the dependencies between different components, the GPUs cannot operate simultaneously, so bubbles exist. Finally, the bottom subfigure illustrates SCPL: in time unit $t_1$, the 1st GPU (Device No. 0) computes the forward path in component 1. At $t_2$, the 2nd GPU takes $\boldsymbol{r}_1^{(i)}$ and $\boldsymbol{r}_1^{(j)}$ as input to perform forward, and the 1st GPU computes the local loss for component 1. At $t_3$, the 3rd GPU takes $\boldsymbol{r}_2^{(i)}$ and $\boldsymbol{r}_2^{(j)}$ as input to perform forward, the 2nd GPU computes the local loss for component 2, and the 1st GPU computes the gradients for the parameters in component 1 via backpropagation. Therefore, different GPUs may compute the parameter gradients in different components simultaneously.

Table 2: The speedup of the training time for SCPL (1, 2, or 4 GPUs) and GPipe (1, 2, or 4 GPUs) using BP of the same batch size as the reference. The actual running minutes of BP are shown in parentheses. We use Transformer as the network and IMDB as the dataset.

| Batch size | 32 | 64 | 128 | 256 | 512 |
|---|---|---|---|---|---|
| BP | 1x (196 min) | 1x (173 min) | 1x (156 min) | 1x (149 min) | 1x (147 min) |
| GPipe (1 GPU) | 0.75x | 0.72x | 0.72x | 0.71x | 0.70x |
| GPipe (2 GPUs) | 1.00x | 0.92x | 0.93x | 0.93x | 0.92x |
| GPipe (4 GPUs) | 1.35x | 1.25x | 1.17x | 1.16x | 1.11x |
| SCPL (1 GPU) | 1.12x | 1.07x | 1.03x | 1.03x | 1.05x |
| SCPL (2 GPUs) | 1.43x | 1.37x | 1.32x | 1.37x | 1.38x |
| SCPL (4 GPUs) | 1.92x | 1.82x | 1.66x | 1.67x | 1.66x |

Table 3: The speedup of the training time for SCPL (1, 2, or 4 GPUs) and GPipe (1, 2, or 4 GPUs) using BP of the same batch size as the reference. The actual running minutes of BP are shown in parentheses. We use VGG as the network and Tiny-Imagenet as the dataset.

| Batch size | 32 | 64 | 128 | 256 | 512 |
|---|---|---|---|---|---|
| BP | 1x (204 min) | 1x (215 min) | 1x (220 min) | 1x (224 min) | 1x (244 min) |
| GPipe (1 GPU) | 0.45x | 0.54x | 0.62x | 0.63x | 0.67x |
| GPipe (2 GPUs) | 0.57x | 0.66x | 0.67x | 0.82x | 0.84x |
| GPipe (4 GPUs) | 0.73x | 0.92x | 1.00x | 1.05x | 1.27x |
| SCPL (1 GPU) | 0.66x | 0.80x | 0.89x | 0.92x | 0.98x |
| SCPL (2 GPUs) | 0.82x | 0.98x | 0.97x | 1.19x | 1.24x |
| SCPL (4 GPUs) | 1.04x | 1.37x | 1.44x | 1.53x | 1.92x |

Referring to Figure 4 in Appendix A.2, we use a profiler to show the utilization of GPUs in a real training job. The profiler shows the GPU usage footprint of NMP and SCPL, which is indeed close to our illustration in Figure 3.

## 4 EXPERIMENTS

We compare SCPL with baselines using different neural networks on both image datasets (CIFAR-10, CIFAR-100, and Tiny-ImageNet) and text datasets (AG's news and IMDB). We test the VGG network and the residual network (ResNet) for the image datasets. We tested LSTM and Transformer for text datasets. Detailed hyperparameter settings are reported in Appendix A and can be found in our released code.

### 4.1 SPEEDUP OF THE EMPIRICAL TRAINING TIME

We compare the training time of SCPL (with 1, 2, or 4 GPUs) with BP and GPipe, a representative MP method. The speedup of a method $m$ is defined as the practical training time of BP divided by the training time of $m$. We do not compare the training time with AL because the released code of AL does not include the implementation of model parallelism.

Using VGG as the network architecture and tiny-ImageNet as the experimental dataset, Table 3 compares the speedup of training time under different batch sizes for SCPL and GPipe (with 1, 2, or 4 GPUs). Similarly, Table 2 shows the speedup when using Transformer as the network architecture and IMDB as the experimental datasets. We use VGG and Transformer because they are representative models for vision tasks and natural language processing tasks.

Here are our observations. First, SCPL indeed accelerates the training time as we have more GPUs. However, probably due to the communication and synchronization overheads, the speedup improves sub-linearly with the number of GPUs. Second, although the training efficiency of GPipe improves when we use more GPUs, the improved ratio is worse than that of SCPL. This is likely because GPipe still suffers from the issue of backward locking, so the bubbles are unavoidable. Additionally,

Table 4: A comparison of the test accuracies (mean ± standard deviation) of different methodologies when using different neural network architectures on IMDB. We highlight the winner among the non-BP methodologies and all models that are non-significantly different from the best models in boldface. We mark a † symbol if the test accuracy of this methodology is higher than that of BP.

|  |  | LSTM | Transformer |
|---|---|---|---|
| BP |  | $89.68 \pm 0.20$ | $87.54 \pm 0.44$ |
| Early Exit |  | $84.34 \pm 0.31$ | $80.24 \pm 0.24$ |
| AL |  | $86.41 \pm 0.61$ | $85.65 \pm 0.77$ |
| SCPL |  | $\mathbf{89.84} \pm 0.10$ † | $\mathbf{89.03} \pm 0.12$ † |

Table 5: A comparison of the test accuracies of different methodologies when using different neural network architectures on Tiny-ImageNet. We follow the same notations used in Table 4.

|  |  | VGG | ResNet |
|---|---|---|---|
| BP |  | $48.30 \pm 0.14$ | $49.71 \pm 0.18$ |
| Early Exit |  | $46 \pm 0.18$ | $40 \pm 0.34$ |
| AL |  | $\mathbf{49.06} \pm 0.14$ † | $44.83 \pm 0.15$ |
| SCPL |  | $\mathbf{48.95} \pm 0.17$ † | $\mathbf{46.87} \pm 0.26$ |

although the original GPipe paper (Huang et al., 2019) reported a speedup from 1.7x to 1.8x when using 4 GPUs, these speedups were obtained by setting the number of micro-batches to an extreme number (32). Later experiments have revealed that when using GPipe with multiple GPUs, the training time can sometimes be longer, compared to traditional BP with a single GPU (Zhang et al., 2023). This observation is consistent with our results. Third, despite involving more parameters during training, SCPL with a single GPU has a training time similar to, and sometimes faster than, BP. It may seem counter-intuitive at first glance because SCPL involves more operations, and a single GPU does not seem to parallelize the computation loading. However, the execution of various computation parts on a GPU is asynchronous, so a larger GPU has the capability to execute multiple kernels simultaneously. Due to the asynchronous property, decoupling the computation means that the update of the first component can initiate while the forward propagation of the remaining network occurs, potentially leading to improved utilization of GPU compute units. However, this favorable scenario is not always guaranteed and relies heavily on factors such as tensor sizes and GPU specifications, which might explain why it is not consistently observed. That being said, this interesting discovery makes SCPL an attractive option even in a single-GPU environment.

Experiments for the training time of SCPL and BP on other networks and other datasets show similar results; see Appendix A.3 for details.

## 4.2 ACCURACY COMPARISON

This section reports comparisons of the test accuracies of models trained by BP, the Early Exit, and AL, a state-of-the-art method for BP decoupling in terms of test accuracy. Early Exit refers to the strategy of assigning a local objective to a component by adding a local auxiliary classifier that outputs a predicted $\hat{y}$ and updating local parameters based on the difference between $\hat{y}$ and the ground-truth target $y$. We don't include GPipe because its accuracy would be the same as BP when the GPipe's micro-batch size equals BP's mini-batch size. We extensively search for the appropriate hyperparameters for each model to ensure fair comparisons. Additionally, we repeat each experiment 5 times and report the average and standard deviation.

Table 4 shows the results on the IMDB, a text classification dataset. We use pre-trained Glove word embeddings (Pennington et al., 2014) of dimensionality 300 in the first layer of the model for both LSTM and Transformer. The simple Early Exit mechanism can be used to learn the relationship between an image and its class. However, the test accuracies of Early Exit are much worse than those of BP. This is likely because Early Exit loses too much information about the input by reducing the representation to fit the labels in each layer. Such greedy behavior may obtain less-optimal

representations. AL yields test accuracies better than Early Exit on both LSTM and Transformer. SCPL performs best among methodologies using local objectives for training. The test accuracies are comparable to and sometimes better than BP. The experimental results on AG's news, another text classification task, are presented in Table 6 in Appendix A.4. The results are similar: Early Exit is the worst, and SCPL produces the best results among local-objective-based learning strategies.

Table 5 gives the results on an image classification dataset, Tiny-ImageNet. Both AL and our proposed SCPL yield test accuracies that are better than those of BP based on the VGG architectures. However, when ResNet is used, BP yields the highest test accuracy. If we compare only the methods that involve BP decomposition, SCPL still performs the best among them when using ResNet as the network architecture. We also tested BP, Early Exit, AL, and SCPL on CIFAR-10 and CIFAR-100. The results, as shown in Table 7 and Table 8, respectively, in Appendix A.4. The results are similar to those on Tiny-ImageNet: SCPL performs best among local objective-based training strategies in all network architectures, but SCPL performs worse than BP when ResNet is used. These results are also consistent with those reported in (Kao & Chen, 2021; Wu et al., 2022).

### 4.2.1 DISCUSSION ON ACCURACY COMPARISON

When BP is used, all parameters are updated to minimize a global objective – the residual between $\hat{y}$ and $y$. On the other hand, methods to decouple end-to-end backpropagation, such as SCPL and AL, are composed of many local objectives, which may differ from the global objective. Therefore, it is surprising that SCPL and AL outperform BP for some network structures. The authors of AL proposed several conjectures to explain this remarkable result. First, projecting the feature vector $x$ and the target $y$ into the same latent space may be helpful. Second, the autoencoder component used in AL may implicitly perform feature extraction and regularization. Third, overparameterization may be helpful for optimization (Arora et al., 2018; Chen & Chen, 2020). However, the first and second conjectures only apply to AL but not to SCPL, but SCPL still yields better accuracies than BP and AL in many cases. Therefore, the above conjectures may not fully explain the success of SCPL. We surmise that, in SCPL, each local component $\ell$ learns to map its input $r_{\ell-1}^{(i)}$ to a better representation $r_{\ell}^{(i)}$, which probably removes noise and outliers. This is similar to the use cases where contrastive learning is applied to learn better representations of input features in an unsupervised manner.

## 5 CONCLUSION, LIMITATION, AND FUTURE WORK

This paper presents SCPL, an innovative yet simple methodology for decoupling the components of the BP process in a neural network. SCPL is applicable to all supervised discriminative models, potentially benefiting the training of various large models. Our experiments on multiple open datasets and popular network architectures demonstrate that SCPL can significantly reduce training time while achieving test accuracies comparable or superior to the traditional backpropagation algorithm. Moreover, SCPL can potentially address issues arising from long-gradient flows in deep neural networks. Compared with AL, a state-of-the-art alternative to backpropagation, SCPL is more flexible because it does not require additional fully connected layers near the output layers. This flexibility makes SCPL a promising substitute for AL and an attractive alternative to backpropagation. We believe that SCPL has the potential to advance the field of deep learning and contribute to the development of model parallelism and more efficient alternatives for end-to-end backpropagation.

SCPL has the following limitations that may deserve further study. First, SCPL involves auxiliary parameters, which can lengthen the training time due to their participation in training and require a larger memory footprint. This issue is particularly prominent in the visual domain of SCPL. It is possible to reduce the size of the feature map of each block by pooling. Second, forward locking still exists in SCPL, so exploring the possibility of further subdividing forward tasks into smaller micro-batches, similar to GPipe, could be a worthwhile direction; details are discussed in the Appendix A.5. Third, while data parallelism does not usually need any changes to the network architecture, model parallelism typically requires some alterations to the network structure, and SCPL is no exception. This limitation may hinder practitioners from using SCPL. Careful packaging may lower the barrier. For example, the `torchpipe` library[2] simplifies the implementation of GPipe. So, another future work is to implement a library for SCPL to provide a better user interface for developers.

---

[2] `https://github.com/kakaobrain/torchgpipe`

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

## A APPENDIX

### A.1 DETAILS OF EXPERIMENTAL SETTINGS

In our vision tasks, both BP and SCPL adopt a cosine learning rate scheduler, starting from an initial learning rate of 10e-3 and decaying to 10e-5. We use Adam as the optimizer. For data augmentation, we refer to the settings in Khosla et al. (2020) on CIFAR-10 and CIFAR-100, where each image undergoes resizing, random cropping, random horizontal flipping, jittering, and random grayscaling to generate two augmented views as inputs. Consequently, the batch size increases from the original $N$ to $2N$. Regarding the batch size, in the accuracy experiments, BP is set to 128, while SCPL is set to 1024. However, in training time measurement experiments, both are tested under 32, 64, 128, 256, and 512. The training epochs are set to 200. In the SCPL configuration, all models use an MLP (multi-layer perceptron) as the projection head, with a structure of $Linear(dim, 512) - ReLU() - Linear(512, 1024)$. Here, $dim$ represents the dimension after flattening the feature map. The temperature parameter $\tau$ is set to 0.1 for all models. Furthermore, in the training time experiments, each component is placed on a separate GPU. The detailed configurations for VGG and ResNet are as follows.

- VGG: It consists of 4 max-pooling layers (MP) and 6 convolutional layers (Conv). Each convolutional layer uses ReLU as the activation function and employs batch normalization (BN). The classifier consists of 2 fully connected layers (FC) with sigmoid as the activation function between the two layers. In our implementation, SCPL splits VGG into 4 components, structured as $component_1 - component_2 - component_3 - component_4$. The $component_1$ and $component_2$ are composed of $[[Conv - BN - ReLU] \times 2 - MP]$. The $component_3$ and $component_4$ are composed of $[[Conv - BN - ReLU] - MP]$. However, an additional classifier is included in $component_4$, resulting in $[component_4 - [FC - sigmoid - FC]]$.

- ResNet: It is an 18-layer residual neural network (ResNet-18) with a linear fully connected layer as the classifier. In our implementation, SCPL splits ResNet into 4 components, structured as $component_1 - component_2 - component_3 - component_4$. The $component_1$ is structured as $[StemBlock - BasicBlock \times 2]$. The $component_2$, $component_3$, and $component_4$ are structured as $[BasicBlock \times 2]$. However, an additional classifier is included in the last $component_4$, resulting in $[component_4 - [FC]]$. The $StemBlock$ includes $[Conv - BN - ReLU]$. The $StemBlock$ is composed of a convolutional layer followed by batch normalization (BN) and Rectified Linear Unit (ReLU) activation. The $BasicBlock$ is constructed with a convolutional layer using the LeakyReLU activation function, another convolutional layer, and a skip connection that adds a fully connected transformation to the input of the $BasicBlock$ module. Finally, the output passes through another LeakyReLU activation.

In NLP tasks, both BP and SCPL use a fixed learning rate of 10e-3 and employ Adam as the optimizer. All texts in the datasets undergo preprocessing steps such as creating word indices, removing stop words, and limiting the maximum text word length $T$, which is a hyperparameter representing the sentence length for each sample. Data augmentation is not utilized, and therefore, the batch size remains at its original value $N$. For the AG's news dataset, the maximum text word length per sample is set to 60, while for IMDB, it is set to 350. The training epochs for both BP and SCPL models are set to 50. Regarding the batch size, we experimented with 16, 32, 64, 128, 256, 384, 512, 768, 1024, 1280, 1536, 1792, 2048, and 4096, and we present the results with the best accuracy in this paper. Additionally, both BP and SCPL models utilize pre-trained Glove word embeddings (Pennington et al., 2014) of dimensionality 300 in the first layer of the model. In the configuration of SCPL, all models by default use an identity function, $f(x) = x$, as the projection head in training. The temperature parameter $\tau$ is set to 0.1 for all models.

Detailed architectures of LSTM and Transformer are as follows.

- LSTM: It consists of 3 bi-LSTM hidden layers (each with a dimensionality of 300) and 1 Glove embedding layer at the beginning of the model. At the end of the model, there are 2 fully connected layers serving as the classifier. The Tanh function is used as the activation function between the two layers. SCPL splits the LSTM model into 4 components, structured as $component_1 - component_2 - component_3 - component_4$. $Component_1$ represents

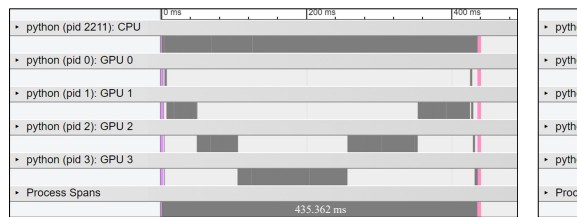 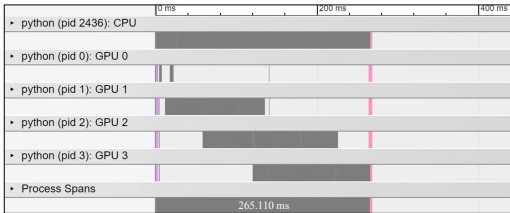

(a) Training LSTM on IMDB (using NMP).  (b) Training LSTM on IMDB (using SCPL).

Figure 4: Visualizing the training job of each device.

the $[GloveEmb]$ layer, while $component_2$, $component_3$, and $component_4$ represent the $[LSTM]$ layers. However, an additional classifier is included in $component_4$, resulting in $[component_4 - [FC - tanh - FC]]$.

- Transformer: It consists of 3 Transformer encoders (each with a dimensionality of 300 and a dropout rate of 0.1) and 1 Glove embedding layer at the beginning of the model. At the end of the model, there are 2 fully connected layers serving as the classifier. The Tanh function is used as the activation function between the two layers. SCPL splits the Transformer model into 4 components, structured as $component_1 - component_2 - component_3 - component_4$. $Component_1$ represents the $[GloveEmb]$ layer, while $component_2$, $component_3$, and $component_4$ represent the $[Transformer]$ layers. However, an additional classifier is included in $component_4$, resulting in $[component_4 - [FC - tanh - FC]]$.

## A.2 Profiling NMP and SCPL

We used PyTorch's profiler to observe the operating periods of the CPU and the GPUs of one iteration. We used 4 GPUs to train an LSTM with 4 layers; each GPU is responsible for the training of one layer.

Figure 4 shows the CPU's working periods and each GPU's working periods when training by NMP and SCPL. The top row shows the CPU's running periods. Since the CPU handles task scheduling, data preprocessing, data management, and some non-parallel computation, the CPU is running throughout the training periods.

When training by NMP (Figure 4(a)), the GPU0 performs forward for layer 1, then GPU1 performs forward for layer 2, then GPU2 performs forward for layer 3, then GPU3 performs forward for layer 4. GPU3 continues to perform backward for layer 4, then GPU2 continues to perform backward for layer 3, then GPU1 continues to perform backward for layer 2, then GPU0 continues to perform backward for layer 1. Finally, the CPU asks all GPUs to update the parameters based on the computed gradients (the red bars). As shown, all the GPUs perform operations sequentially, causing backward locking, so many bubbles exist among the dependent tasks. The total training time for this iteration is 435.362 ms.

When training by SCPL (Figure 4(b)), the operations on different GPUs may overlap. In particular, when GPU0 finishes the forward for layer 1, the following operations may occur simultaneously: backward for layer 1 (on GPU0) and forward for layer 2 (on GPU1). Similarly, when GPU1 finishes the forward for layer 2, backward for layer 2 (on GPU1) and forward for layer 3 (on GPU2) may occur simultaneously. After GPU2 finishes the forward, backward for layer 3 (on GPU2) and forward for layer 4 (on GPU3) may occur concurrently. Finally, GPU3 performs the backward for layer 4, and then the CPU issues an update command for all GPUs (the red bars). Since many bubbles are removed, the total training time for this iteration is reduced to 265.110 ms.

## A.3 More comparisons on empirical training time

This section shows the empirical training time per epoch for NLP and vision tasks using famous network architectures.

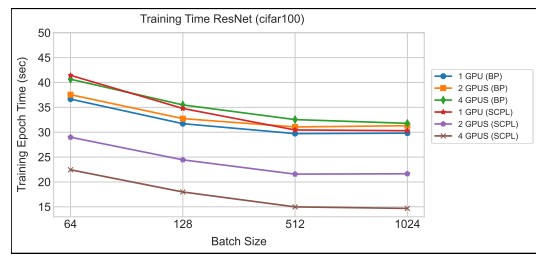

Figure 5: Empirical training time per epoch using ResNet architecture on CIFAR-100.

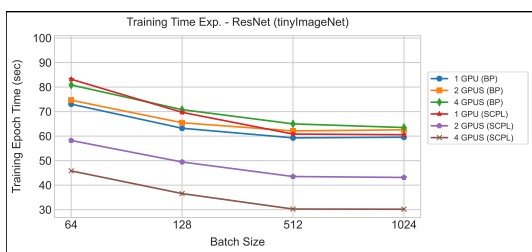

Figure 6: Empirical training time per epoch using ResNet architecture on tiny-ImageNet.

For the NLP tasks, we apply the LSTM and Transformer networks; the experimental datasets include AGNews and IMDB. Regarding vision tasks, we select the VGG network and ResNet, and the dataset includes CIFAR-100 and tiny-ImageNet.

The results are shown in Figures 5, 6, 8, 8, 9, 10, 11, and 12. When using 4 GPUs, SCPL is approximately 2 times faster than BP on vision tasks and approximately 1.6 times faster on NLP tasks.

### A.4 More comparisons on test accuracies

Table 6: A comparison of the test accuracies of different methodologies when using different neural network architectures on AG's news. We follow the same notations used in Table 4.

|  | LSTM | Transformer |
|---|---|---|
| BP | $91.97 \pm 0.19$ | $91.27 \pm 0.18$ |
| Early Exit | $85.91 \pm 0.11$ | $85.79 \pm 0.43$ |
| AL | $91.53 \pm 0.20$ | $\mathbf{91.17} \pm 0.43$ |
| SCPL | $\mathbf{92.12} \pm 0.04$ † | $\mathbf{91.64} \pm 0.23$ † |

This section shows the test accuracies for NLP and vision tasks using famous network architectures.

Figure 6 shows the test accuracies of LSTM and Transformer on AG's news when these models are trained by BP, Early Exit, AL, and SCPL. We report the mean and standard deviation of 5 trials.

Similarly, we also report the results on CIFAR-10 and CIFAR-100, using the vanilla convolutional neural network (Vanilla ConvNet), VGG, and ResNet as the network structures. The results are shown in Table 7 and Table 8.

In general, SCPL consistently outperforms other local-objective-based learning strategies in the experimented datasets and different network architectures.

### A.5 SCPL vs. GPipe

SCPL and GPipe share an architectural similarity: they both rely on pipelining to realize model parallelism and enhance the training throughput. However, they adopt distinct strategies to address

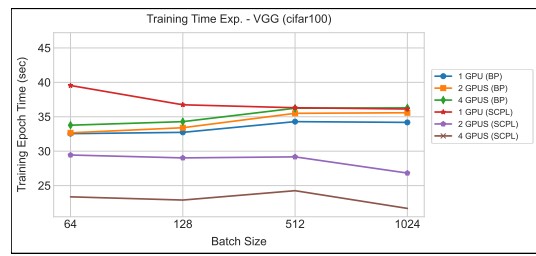

Figure 7: Empirical training time per epoch using VGG architecture on CIFAR-100.

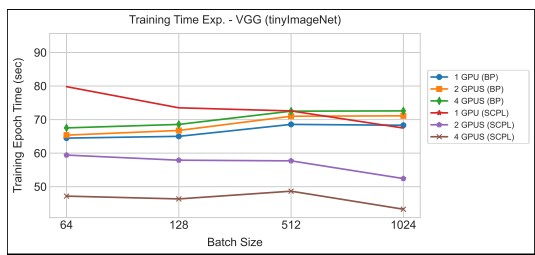

Figure 8: Empirical training time per epoch using VGG architecture on tiny-ImageNet.

the challenges posed by forward and backward locking. SCPL and GPipe can be integrated to further improve training throughput.

SCPL focuses on mitigating backward locking, where the sequential dependency of gradient calculations across layers impedes parallelism. SCPL introduces a local objective for each component. These local objectives serve to disentangle the gradient computation process, allowing greater concurrency and minimizing the impact of backward locking. Referring to Figure 3 and the top subfigure in Figure 13, the backward pass in different components can be computed simultaneously in different GPUs.

GPipe tackles forward locking, a phenomenon in which the forward operation of a layer must wait for the completion of the forward operations in the earlier layers. GPipe alleviates the constraint by subdividing traditional mini-batches into micro-batches, allowing for an overlap of computations between the forward passes of different layers. This approach mitigates the impact of forward locking. Referring to the middle subfigure in Figure 13, each mini-batch is further divided into 3 micro-batches. Particularly, letting FW$\ell$ refer to the forward operations of a mini-batch at layer $\ell$, we use $F_\ell^1$, $F_\ell^2$, and $F_\ell^3$ to refer to the forward pass of the three micro-batches in this layer. In this setting, once a GPU$i$ finishes the computation of $F_\ell^1$, the GPU$(i + 1)$ can continue to execute $F_{\ell+1}^1$, and the GPU$i$ operates $F_\ell^2$ simultaneously. As a result, it is possible to execute the forward passes at different layers simultaneously.

Given their complementary strengths in addressing forward and backward locking, it is possible to integrate SCPL and GPipe. Such an integration could potentially yield a hybrid approach that capitalizes on the benefits of both methodologies. By subdividing mini-batches and concurrently

Table 7: A comparison of the test accuracies of different methodologies when using different neural network architectures on CIFAR-10. We follow the same notations used in Table 4.

|  | Vanilla ConvNet | VGG | ResNet |
|---|---|---|---|
| BP | $86.85 \pm 0.57$ | $93.02 \pm 0.03$ | $93.95 \pm 0.11$ |
| Early Exit | $83.16 \pm 0.33$ | $91.28 \pm 0.15$ | $89.63 \pm 0.34$ |
| AL | $\mathbf{86.98 \pm 0.24}$ † | $\mathbf{93.22 \pm 0.12}$ † | $91.33 \pm 0.09$ |
| SCPL | $\mathbf{86.98 \pm 0.33}$ † | $\mathbf{93.42 \pm 0.11}$ † | $\mathbf{92.78 \pm 0.11}$ |

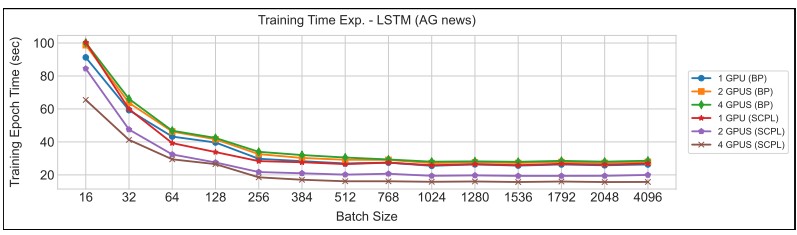

Figure 9: Empirical training time per epoch using LSTM architecture on AGNews.

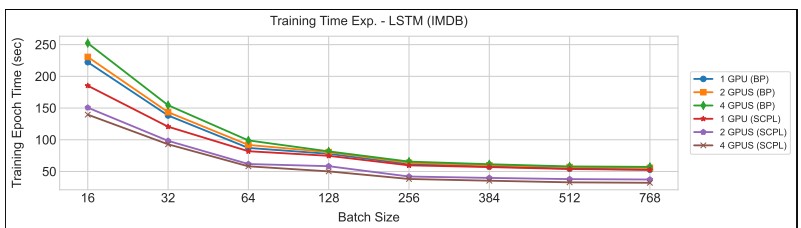

Figure 10: Empirical training time per epoch using LSTM architecture on IMDB.

designing local objectives, a harmonized pipeline structure may offer a solution to enhance training efficiency for large-scale neural network models.

The bottom subfigure of Figure 13 illustrates the integration of both SCPL and GPipe. Each mini-batch is divided into 3 micro-batches, so forward locking can be partially addressed, as demonstrated in $t_1$ to $t_6$. Additionally, since we allocate the local objective for each component using SCPL, each GPU can compute the local objective for each component and further compute the local gradients without waiting for the gradient information computed by other GPUs. In this example, the integration needs 22 time steps to complete one iteration of forward, backward, and parameter update, whereas SCPL and GPipe need 24 time steps and 31 time steps, respectively.

## A.6  PSEUDO CODE

To help understand the details of SCPL, here are pseudocodes for local supervised contrastive losses (Algorithm 1) and SCPL without pipelining (Algorithm 2).

Table 8: A comparison of the test accuracies of different methodologies when using different neural network architectures on CIFAR-100. We follow the same notations used in Table 4.

|  | Vanilla ConvNet | VGG | ResNet |
|---|---|---|---|
| BP | $58.68 \pm 0.13$ | $72.58 \pm 0.39$ | $73.59 \pm 0.11$ |
| Early Exit | $50.64 \pm 0.44$ | $71.11 \pm 0.95$ | $64.48 \pm 0.41$ |
| AL | $53.06 \pm 0.15$ | $72.43 \pm 0.27$ | $67.53 \pm 0.32$ |
| SCPL | $\mathbf{59.63} \pm 0.37$ † | $\mathbf{73.14} \pm 0.30$ † | $\mathbf{70.41} \pm 0.27$ |

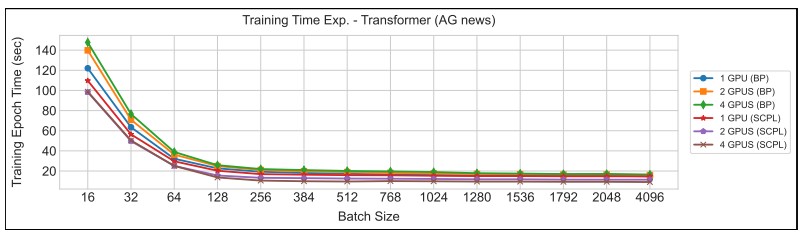

Figure 11: Empirical training time per epoch using Transformer architecture on AGNews.

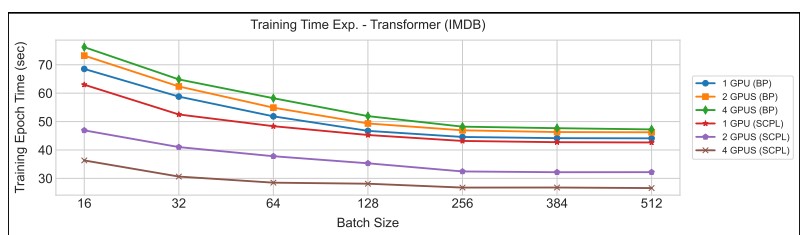

Figure 12: Empirical training time per epoch using Transformer architecture on IMDB.

```python
import torch
import torch.nn as nn

class SupConLoss(nn.Module):
    def __init__(self, dim):
        super.__init__()
        self.linear = nn.Sequential(nn.Linear(dim, 512), nn.ReLU(), nn.Linear(512, 1024))
        self.temperature = 0.1

    def forward(self, x, label):
        x = self.linear(x)
        x = nn.functional.normalize(x)
        label = label.view(-1, 1)
        bsz = label.shape[0]
        mask = torch.eq(label, label.T).float()
        anchor_mask = torch.scatter(torch.ones_like(mask), 1, torch.arange(bsz).view(-1, 1), 0)
        logits = torch.div(torch.mm(x, x.T), self.temperature) deno = torch.exp(logits) * anchor_mask
        prob = logits - torch.log(deno.sum(1, keepdim=True))
        loss = -(anchor_mask * mask * prob).sum(1) / mask.sum()
        return loss.view(1, bsz).mean()
```

Algorithm 1: PyTorch-like pseudocode for $L^{sc}$

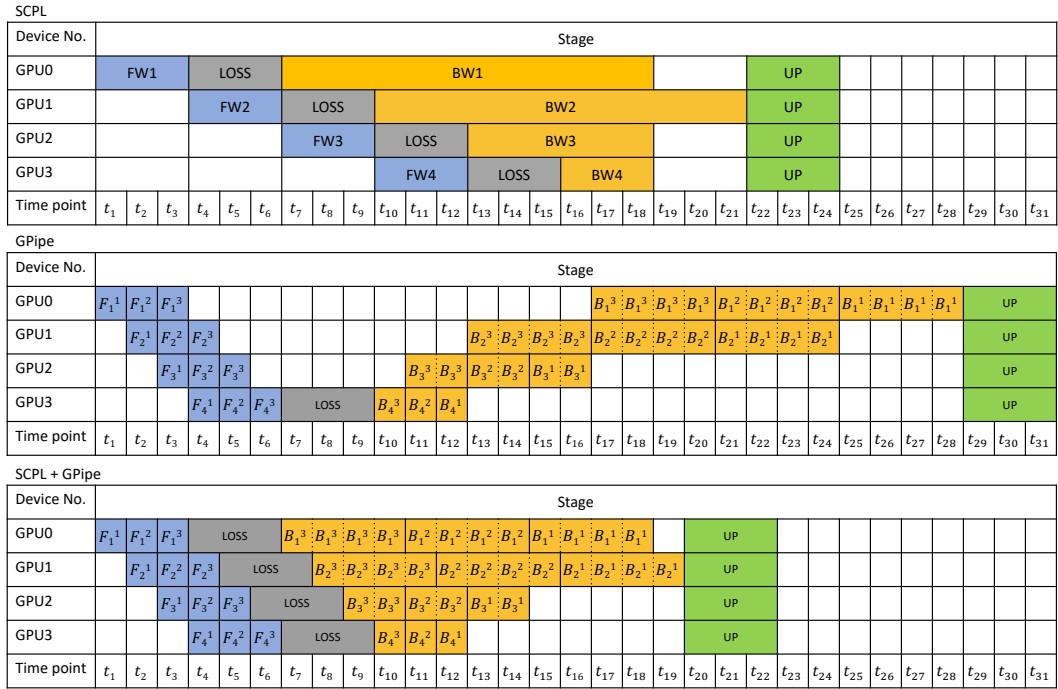

Figure 13: A comparison of SCPL, GPipe, and an integration of both.

```python
import torch
import torch.nn as nn

# A simple 3-layer CNN example for SCPL architecture.
class CNN_SCPL(nn.Module):
    def __init__(self, dim):
        super.__init__()
        CNNs = [ ]
        losses = [ ]
        channels = [3, 128, 256, 512] self.shape = 32
        for i in range(3):
            CNNs.append(nn.Sequential(nn.Conv2d(channels[i], channels[i+1], padding=1), nn.ReLU()))
            losses.append(SupConLoss(self.shape*self.shape*channels[i+1]))
        self.CNN = nn.ModuleList(CNNs)
        self.loss = nn.ModuleList(losses)
        self.fc = nn.Sequential(flatten(), nn.Linear(self.shape*self.shape*channels[-1], 10))
        self.ce = nn.CrossEntropyLoss()

    def forward(self, x, label): loss = 0
        for i in range(3):
            # .detach() prevents a gradient flows to neighboring layer
            x = self.CNN[i](x.detach())
            if self.training:
                loss += self.loss[i](x, label)
        y = self.fc(x.detach())
        if self.training:
            loss += self.ce(y, label)
            return loss
        return y
```

Algorithm 2: PyTorch-like pseudocode for SCPL without pipelining

