# OpenReview forum: "Enabling Model Parallelism for Neural Networks Based on Decoupled Supervised Contrastive Learning"
_ICLR.cc/2024/Conference — ICLR 2024 Conference Withdrawn Submission_

### Official Review · Reviewer_3oSN · 2023-10-24

**Soundness:** 3 good
**Presentation:** 2 fair
**Contribution:** 2 fair
**Rating:** 5
**Confidence:** 4

**Summary:**

The authors present SCLP for model parallelism in contrastive learning tasks. In that, the authors overlap the computations of multiple neural network layers by shortening the backpropagation path through the entire network, by layer-wise learning problems. Instead of propagating the gradient through all layers, each layer is learning a local loss objective, thereby updating its local weights. In an experimental evaluation with up to 4 GPUs, on multiple datasets (IMDB, Tiny-ImageNet, CIFAR-10/100) and different architectures (LSTM, Transformer, Pyramid CNNs and ResNets) the authors claim improved predictive performance and speed-up compared to vanilla BP and GPipe.

**Strengths:**

- Clear and understandable writing
     * Especially good figures supporting the text
- Straight-forward idea to better overlap computations with a promise for high gains
- Already extensive experimental evaluations on different datasets and architectures
- Solid experimental results

**Weaknesses:**

- Applicability of the approach is limited to contrastive learning problems and does not generalize well to other learning objectives, could you generalize it?
- The degree of parallelization is limited to the number of layers in a neural network, which become increasingly more shallow
- The provided repository seems to be leaking the authors' identity by having a link to its non-anonymous counterpart on GitHub
- No experimentation regarding SCLP's behavior with different number of layers, seems fundamental to the applicability to arbitrary problems
- Possibly use sub-figures, e.g. Fig. 1. and 2.
- It is unclear how VGG (which?) has been split onto 1, 2 and 4 GPUs respectively, there are more layers than the previously stated 4-layer network in Fig. 2, same is true for ResNet (which?), should definitely be part of the main textual body
- Related work could be extended, especially data-parallel, model-parallel as well as the decoupling ideas have been previously published in the following works, but not discussed:
    * Nøkland, A. (2016). Direct feedback alignment provides learning in deep neural networks. Advances in neural information processing systems, 29.
    * Hinton, G. (2022). The forward-forward algorithm: Some preliminary investigations. arXiv preprint arXiv:2212.13345.
    * Flügel, K., Coquelin, D., Weiel, M., Debus, C., Streit, A., & Götz, M. (2023). Feed-Forward Optimization With Delayed Feedback for Neural Networks. arXiv preprint arXiv:2304.13372.
    * Dellaferrera, G., & Kreiman, G. (2022, June). Error-driven input modulation: solving the credit assignment problem without a backward pass. In International Conference on Machine Learning (pp. 4937-4955). PMLR
- Does not separate own contributions well enough from related work, e.g. the supervised contrastive loss was introduced in "Supervised Contrastive Learning" by Khosla et al. and was only modified to take the layer output instead of the network output
- The optimization issues (other than the impact on parallelization/performance) caused by long gradients flows are mentioned (e.g. in Section 3.2) but never really explained. The same applies for the disadvantages of un-supervised training. This makes Table 1 less useful than it could be
 - Early Exit is only briefly introduced in Section 4.2, which is a bit late since it's already referred to in the abstract and introduction, and it is missing a citation
 - A greater speedup than GPipe was to be expected as already outlined in Figure 13, a comparison (both in terms of speedup and accuracy) to higher-throughput approaches like PipeDream would have been interesting (while the asynchronous updates of PipeDream may impact its final accuracy, SCPL can also result in lower accuracy than backpropagation as shown for ResNet in Table 5).

**Questions:**

- It is not quite clear to the reviewer why Eq. 3 would be correct. It seem to be that the global objective is only $\mathcal{L}(B)=\sum_{l=1}^{H}\mathcal{L}_{l}^{SC}(B)$.

It is not clear why the backwards objective with the full backpropagation chain, i.e., $\sum_{\forall i\in B}^{}\mathcal{L}_{i}^{OUT}$, would also be optimized. In the reviewers understanding, there would be at most an optimization of an approximation of this function.

- "(we ignore the communication cost between GPUs)" p. 6 - communication is one of the major bottlenecks in parallel neural network training, why is this a valid simplification?
- It is unclear which SCPL configurations have been used in Tables 4 and 5. Is there a decrease in predictive performance?
- In section 4.2, the paper states that "[GPipe's] accuracy would be the same as BP when the GPipe’s micro-batch size equals BP’s mini-batch size". Shouldn't this hold independent of the micro-batch size since GPipe uses synchronous updates and should thus always be equivalent to BP? If not, wouldn't it be interesting to include GPipe's accuracy in Tables 4 and 5 for the micro-batch size chosen for Tables 2 and 3?

---

### Official Review · Reviewer_sXbp · 2023-10-29

**Soundness:** 3 good
**Presentation:** 3 good
**Contribution:** 2 fair
**Rating:** 3
**Confidence:** 4

**Summary:**

The paper proposes a paralliesation of the supervised contrastive learning (SCL) taking advantage of the fact that it allows independent updates of hidden layers, which is not possible with the classical backpropagation.  Empirical evaluation shows that parallelisation offers a speed up while SCL itself improves the accuracy of the trained model.

**Strengths:**

The paper examines and evaluates the potential of supervised contrastive learning training for the purpose of additional parallelisation in neural networks.  The results shows that the proposed supervised contastive parallel method is faster than comparable baselines while (remarkably) improving the accuracy of the model.

The paper is well written.

**Weaknesses:**

There is no new training algorithm proposed - the proposed method just takes advantage of the fact that contrastive learning offers layer-wise loss functions that can be trained in parallel.  Provided empirical evaluation of the speed up is probably valuable, but not sure this conference is the right target.

For this conference, the more interesting issue is why and when SCL produces improved generalisation.  The implementation of SCL as SCPL (P for parallel) seems like something that is of more interest to systems rather than learning representation crowd.  And this paper does not have much to say about the former (other than showing empirical results on two datasets) and focuses on the latter.

I wonder if the comparison of either the speed-up or the accuracy of the propose parallelisation against standard BP is fair (see questions below).

**Questions:**

As I understand, SCPL requires augmentation that doubles the amount of data that is used for training.  So...were the BP and GPipe networks in Tables 2 and 3 trained on the augmented dataset or not?  Because BP does not require augmented examples to train, whereas SCPL does.  I understand training on same augmentation gives a fair comparison for the accuracy, but that takes for granted that we want to train on augmented data.  Perhaps there are cases where augmentation doesn't help, and so BP could be trained on a smaller dataset, and so (potentially) could be faster?

I am aware this is almost a reverse argument, but when it comes to accuracy results - has BP been trained on augmented data?  Because if not, then the accuracy comparison is unfair, given SCPL was effectively trained on augmented dataset (while BP was not).

Why in the accuracy experiments, the batch size is set to 128 for BP, and 1024 in SCPL (as I understand from note in A1)?  Batch size of 1024 wasn't even used in the speed up comparison, and from Table 2 it is evident that larger batch size is inversely correlated with the speed up.  And so, is SCPL trained in the "slower" mode to get a better accuracy here?  The only justification for not evaluating accuracy across different batch sizes would be if the batch size choice does not much affect the accuracy results...but if that's the case, why the need to use different batch sizes for different methods?  If batch size has an effect on accuracy of SCPL, surely results in Tables 4 and 5 need to show performance over different choice of batch size, like in Tables 2 and 3.

---

### Official Review · Reviewer_KKRG · 2023-10-31

**Soundness:** 2 fair
**Presentation:** 3 good
**Contribution:** 3 good
**Rating:** 5
**Confidence:** 4

**Summary:**

The present paper tackles the so-called "backward locking problem" of the backpropagation (BP) algorithm, i.e. the sequentiality of the backpropagation algorithm along with a global objective locks the weight update of a given module until an upstream error signal reaches it. The proposed solution, which the authors call "supervised contrastive parallel learning" (SCPL), is the local counterpart of the SimCLR algorithm [1] combined with supervised contrastive learning (SCL)  [2] to use label information to define the positive / negative pairs of data samples. The core algorithmic ingredient is the following: instead of having a single encoder $f$ and projection head $g$ which are optimized globally to optimize the InfoNCE loss $\ell$ end-to-end with BP, local encoders and projection heads $f^{(i)}$ and $g^{(i)}$ are optimized *locally* with local InfoNCE losses $\ell^{(i)}$. This way, parameter gradients of layer $i$ can be computed before those of layer $i+1$ and possibly even before the forward pass has ended, thereby allowing for efficient pipelining and reduction of latency thereof. It is experimentally shown that SCPL outperforms, or at least performs on par, across almost all experimental settings (vision and NLP models and data) with BP, backward-unlocked (e.g. early exit, associated learning) and forward-unlocked (e.g. GPipe) gradient computation techniques, both in terms of latency / speedup (with respect to vanilla BP) and performance.

More precisely:
- Section 2 introduces related work on model parallelism (e.g. associated learning, GPipe), backward-unlocking with local self-supervised objectives (e.g. LoCo, GIM) and motivates the scope of comparison chosen for the paper.

- Section 3 introduces the methodology previously described, namely: contrastive learning and its supervised counterpart (3.1), the SCPL algorithm (Fig. 2) in terms of its local losses (3.2), how gradients are blocked (3.3) and of the resulting pipelining (Fig. 3, 3.4). Importantly, the naive model parallelization (NMP) baseline employing vanilla (backward-locked) BP is introduced and serves as a reference for the experiments.

- Section 4 shows the experimental results obtained, analyzing separately the training speedup obtained (4.1) and the accuracy comparison (4.2).

- **Training speedup** (4.1): SCPL is compared against GPipe and vanilla BP across different images (CIFAR10, CIFAR100, TinyImageNet) and text (AG's news and IMDB) datasets with respective architectures (VGG and Transformer) for various number of devices and batch sizes (Tables 2 and 3). The key observations for the training speedup analysis is that SCPL indeeds benefits from GPU parallelization (the higher the number of devices, the higher the speedup), the speedup obtained with respect to BP with SCPL is higher than that achieved by GPipe and finally SCPL is still higher than BP even when using a single device because of the asynchronicity of the kernel executions.

- **Accuracy comparison** (4.2): SCPL is compared against AL, Early Exit (two *supervised* technique to unlock the backward pass) and vanilla BP on text (Table 4) and image tasks (Table 5) with respective architectures (LSTM & Transformer on the one hand, VGG and ResNet on the other hand). The main takeaways of this analysis are that SCPL is the best technique compared to AL and Early Exit, and outperforms (or performs on par with) vanilla BP except on ResNets.



[1] *Ting Chen, Simon Kornblith, Mohammad Norouzi, and Geoffrey Hinton. A simple framework for contrastive learning of visual representations. In International conference on machine learning, pp. 1597–1607. PMLR, 2020*

[2] *Prannay Khosla, Piotr Teterwak, Chen Wang, Aaron Sarna, Yonglong Tian, Phillip Isola, Aaron Maschinot, Ce Liu, and Dilip Krishnan. Supervised contrastive learning. Advances in Neural Information Processing Systems, 33:18661–18673, 2020.*

**Strengths:**

- The paper is clearly written and the core idea is clear.
- Experiments are thoroughly described and experimental results are clearly presented.
- Evaluation of SCPL is done across many datasets and models.
- SCPL effectively benefits from model parallelization and it effectively works in terms of resulting performance.
- SCPL results in a better performance and /or (?) training time than some techniques alleviating forward-locking (GPipe) and backward-locking (Early exit, AL).

**Weaknesses:**

- The choice of the algorithmic baselines, both for training time and accuracy, are poorly justified and not very convincing (see my questions / remarks below). For instance, I really don't understand **why SCPL was not compared against LoCo and GIM** beyond the burden of coding a parallel implementation of these algorithms.
- The comparisons made in the experimental section do not seem to be the most relevant. For instance:
   - **Why comparing SCPL (designed to unlock the backward pass) against GPipe (designed to unlock the forward pass)**? These two algorithms are quite orthogonal, and it is explicitly acknowledged by the authors in their conclusion and appendix by highlighting of these two algorithms could be combined to unlock both the forward and backward pass.
   - **Why disentangling the training time and accuracy analysis?** We have no idea how well the resulting models used for Tables 2 and 3 perform, neither do we know about the training time of the models used in Table 4 and 5.

- As a minor remark: the paper is not self-contained as it does not sufficiently describe the GPipe and Early Exit algorithm.

**Questions:**

- A minor (compared to my other points) yet important remark. You write p. 2:  *"Therefore, the various methods that attempt to find alternative BP methods could potentially **achieve model parallelism**. Studies in this line include target propagation (TP) (Lee et al., 2015; Meulemans et al., 2020; Manchev & Spratling, 2020; Bengio, 2014), gradient predic- tion (Jaderberg et al., 2017), and local objective assignments (Wu et al., 2022; Kao & Chen, 2021)*". However, none of the approaches you mention for TP solve the backward locking problem: targets in TP are recursively computed from the top to the bottom of the architecture, mimicking how BP gradients flow. So I could recommend to remove the references to TP as they seem irrelevant to me.

- p.3 you write: "*Greedy InfoMax (GIM) (Löwe et al., 2019) and LoCo (Xiong et al., 2020). While these studies localize the losses, they mainly focus on improving model accuracy and modularity but pay little attention to model parallelism*". Until here, I agree in the sense that while these approaches theoretically allow for backward pass parallelization, the authors of these studies may not have coded up an actual parallel implementation of their algorithms and rigorously measured the resulting speedup compared to BP. However I strongly disagree on the last point you raise: *"Additionally, **these works are primarily unsupervised (self-supervised), so their results are not directly comparable to those of our work**.*" However, **these approaches are no more self-supervised than SimCLR** which SCPL is based on. In the same way as you train a classifier on top of the last layer in SCPL, any self-supervised approaches (like the ones you excluded from your study) also evaluates the quality of the representations by learning a classifier on top of it. So the only justification left as to why you may not have considered these algorithms is (understandably so) because they did not release a parallel implementation of their algorithm (as you acknowledge for the AL algorithm). While I do bear in mind the engineering burden of re-implementing a parallel implementation of existing algorithms, **it is hard to tell from the present study if SCPL performs better (in terms of training time and performance) than existing backward locking-free algorithms**. Also, as you acknowledge in your conclusion, for lack of "careful packaging" of SCPL which "may hinder practitioners from using SCPL", follow-up works may choose, for the exact same reasons as yours, not to compare novel backward unlocked training techniques against yours.
- Rather than decoupling the accuracy and training time analysis on different models, why didn't you **simultaneously** evaluate the performance *versus* the training time? It would clarify a lot to have such 2D plot for *all* the tasks considered.
- Instead of comparing SPCL with GPipe which solve orthogonal problems (respectively backward and forward locking) to assess training time, **why not simply comparing SCPL with its global counterpart supervised contrastive learning (SCL)** and with  a **fixed wall-clock time**?
- In Table 4 and 5, you evaluate several local training techniques, including those for which you don't have a parallel implementation. Why not evaluating GIM and LoCo here?
- What are precisely the early exit and and AL algorithm? Why aren't there any citation for early exit? Is this related to Belilovsky et al (2019)?
- Coming back on the reasons why BP still outperforms SPCL on ResNets: isn't it simply because SPCL discards residual connections? Assume layer i is connected to layer j with $j > i$ by a residual connection. Even when using SPCL, if you want to take into account this residual connection, the backward pass on $\ell^{(i)}$ inside layer i is locked until the backward pass on $\ell^{(j)}$ inside layer j has been performed. Alternatively, the backward pass of layer $i$ can be performed before that of layer $j$, however it doesn't take into account the residual connection. What did you do in your own experiments?

---

### Official Review · Reviewer_iF7K · 2023-11-02

**Soundness:** 3 good
**Presentation:** 4 excellent
**Contribution:** 2 fair
**Rating:** 3
**Confidence:** 2

**Summary:**

The authors introduce the Supervised Contrastive Parallel Learning (SCPL) method aimed at resolving the "backward locking" problem encountered during the training of substantial neural networks across multiple GPUs. This challenge limits the parallelism, necessitating a sequential transfer of gradient information between GPUs. SCPL's primary innovation is the decoupling of gradient flow via supervised contrastive learning. By giving each segment of the network a distinct local objective, SCPL allows for simultaneous gradient calculations without relying on GPU dependencies. The paper's empirical studies, spanning multiple neural network architectures, reveal that SCPL enhances training speed without compromising accuracy in comparison to conventional techniques. Overall, the work signifies a promising stride in the realm of model parallelism for large-scale neural networks.

**Strengths:**

1. The paper provides a thorough explanation, making the proposed method reproducible.
2. Excellent visualization techniques are employed to illustrate the algorithms, facilitating understanding.
3. Merging local gradient computations with self-supervised learning stands out as a novel approach in this domain.

**Weaknesses:**

1. The description surrounding gradient length appears overly assertive. It might be more accurate to present it as $O(H/N)$, relative to the total layer count $H$ and the number of local losses $N$ employed. This observation can be further corroborated by the authors' implementation details.
2. The paper does not delve into the influence of $N$—specifically, the count of local losses used and their respective placements—on the system's overall performance. Investigating this could provide more comprehensive insights.
3. Although the paper introduces a synchronized pipeline parallel algorithm, it seems the "bubble" issue persists in the suggested method. For a rounded perspective, it would be beneficial if comparisons concerning accuracy with asynchronous pipeline parallel algorithms (like PipeDream) were provided. This would further highlight SCPL's potential advantages over asynchronous counterparts.

**Questions:**

1. Is there a mistake in in paragraph 1 in Section 2.2, which should be 'the author of AL' ?
2. Also see weaknesses part.